# The Role of Genetics in the Management of Heart Failure Patients

**DOI:** 10.3390/ijms242015221

**Published:** 2023-10-16

**Authors:** Gianpaolo Palmieri, Maria Francesca D’Ambrosio, Michele Correale, Natale Daniele Brunetti, Rosa Santacroce, Massimo Iacoviello, Maurizio Margaglione

**Affiliations:** 1School of Cardiology, Department of Medical and Surgical Sciences, University of Foggia, 70122 Foggia, Italy; gianpaolo.palmieri@unifg.it (G.P.); michele.correale@libero.it (M.C.); natale.brunetti@unifg.it (N.D.B.); 2Medical Genetics, Department of Clinical and Experimental Medicine, University of Foggia, 70122 Foggia, Italy; maria.dambrosio@unifg.it (M.F.D.); rosa.santacroce@unifg.it (R.S.); maurizio.margaglione@unifg.it (M.M.); 3University Cardiology Unit, Polyclinic Hospital of Bari, 70124 Bari, Italy

**Keywords:** heart failure, genetics, diagnosis, prognosis, therapy

## Abstract

Over the last decades, the relevance of genetics in cardiovascular diseases has expanded, especially in the context of cardiomyopathies. Its relevance extends to the management of patients diagnosed with heart failure (HF), given its capacity to provide invaluable insights into the etiology of cardiomyopathies and identify individuals at a heightened risk of poor outcomes. Notably, the identification of an etiological genetic variant necessitates a comprehensive evaluation of the family lineage of the affected patients. In the future, these genetic variants hold potential as therapeutic targets with the capability to modify gene expression. In this complex setting, collaboration among cardiologists, specifically those specializing in cardiomyopathies and HF, and geneticists becomes paramount to improving individual and family health outcomes, as well as therapeutic clinical results. This review is intended to offer geneticists and cardiologists an updated perspective on the value of genetic research in HF and its implications in clinical practice.

## 1. Introduction

Over recent decades, genetics has taken on an increasingly prominent role in the field of cardiovascular diseases, with a specific emphasis on the management of cardiomyopathies. Notably, an increasing number of identified mutations, which affect a range of genes encoding myocardial proteins, have been pinpointed as causative factors for the onset of cardiomyopathies [1]. Such genetic variants carry multiple clinical connotations [2]. From a diagnostic perspective, their presence necessitates screening of the proband’s family members [1,2,3] to assess cosegregation and transmission, and the potential manifestation of cardiomyopathy among family members. Additionally, certain genetic variants hold significant prognostic relevance [4,5], playing a role in determining therapeutic strategies for patients. In the near future, these genetic variants may serve as focal points for gene therapies.

Given this evolving landscape, the emergence of cardiomyopathy specialists and dedicated outpatient clinics is becoming more prevalent. Yet, it is vital to highlight that the diagnosis of cardiomyopathy in numerous patients is intrinsically linked to that of a “de novo” heart failure (HF) [6]. In fact, cardiomyopathies stand as one of the leading causes of HF, encompassing cases with both reduced and preserved ejection fraction. Consequently, HF specialists should be increasingly aware of the clinical implications related to the genetics of cardiomyopathies, especially in terms of identifying genetic variants that influence the onset and progression of these conditions.

Given the intricacies of this domain, patient management necessitates a multidisciplinary methodology [7,8,9]. Collaboration among cardiologists, with a distinct emphasis on those specializing in cardiomyopathies and HF, as well as geneticists, is paramount to optimizing strategies for individuals, their families, and therapeutic clinical interventions. This review endeavors to present geneticists and cardiologists with an updated perspective on the significance of genetic research in HF and its implications in clinical practice.

## 2. Advancements and Applications of Genetics in Cardiovascular Disease

It has been slightly over 20 years since the completion of the Human Genome Project on 14 April 2003 [10]. Thanks to advancements in scientific investigative techniques, we now possess the ability to analyze and interpret data derived from the genome, including complete DNA sequences and the functional interactions between various genes. The introduction of targeted therapy [11,12], or “personalized medicine”, aiming to combine established clinical-pathological indices with cutting-edge molecular profiling [13], emerged from the insights gained through the Human Genome Project. This evolution enables the creation of individualized diagnostic, prognostic, and therapeutic strategies tailored to each patient. Currently, an array of genome-wide platforms is available, which facilitates an exploration of the genome’s influence on the phenotypic and clinical presentation of diseases [14].

In light of these developments, there is an augmented imperative to evaluate individual risk factors and nuances among individuals. This is achieved primarily through the development and computation of genomic risk scores (GRSs) [15,16], which enhance precision, efficacy, and predictability concerning potential diseases. Identifying individuals at heightened risk for prevalent chronic diseases paves the way for targeted preventive measures, such as increased health monitoring, early therapeutic interventions, lifestyle modifications [17,18], and specialized diagnostic procedures. Furthermore, when combined with risk scores, genetic data can predict a patient’s response to therapy and potential side effects from certain medications. This insight empowers clinicians to select the most appropriate pharmacological treatment and dosage tailored to the patient’s needs.

Managing a patient with an inherited cardiovascular disease necessitates consideration of the following crucial factors:Assessing transmission modes: discerning the mode of cardiomyopathy transmission is critical, with a need to determine whether it is monogenic or polygenic [19].Identifying at-risk family members: recognizing those family members potentially at risk [20,21] of developing the same cardiomyopathy is vital, along with determining those in need of prolonged monitoring [22].Establishing genetic counseling and support: genetic counseling, complemented by psychological support, is essential [23]. In certain cases, initiation during childhood might be warranted.Predicting adverse events: clinical management decisions often hinge on the ability to predict potential adverse outcomes linked to a specific disease.Providing prognostic information: offering insights about the disease’s potential severity and progression trajectory is pivotal for both the patient and their family [20].Personalizing therapy: treatment strategies often need tailoring based on the specific mutations identified or the potential application of gene therapy methods [24].

Genetics plays a fundamental role in this broad spectrum of pathologies due to the social, economic, and familial burden associated with the condition [25]. When delivering care to an entire family or a set of individuals with a specific genetic variant or phenotype, it is essential to prioritize early risk stratification. This approach is particularly relevant for pinpointing individuals or groups at risk of sudden cardiac death (SCD), symptoms signaling the advancement of HF, or the need for heart transplantation. Some patients might only come to the forefront if the focus remains strictly on treating the proband (the first family member identified by a clinician due to a potential genetic disorder, or the one undergoing genetic testing to confirm the diagnosis). Consequently, it is crucial to delve into genetics on a case-by-case, family-by-family basis. This ensures a comprehensive understanding of each patient’s genetic backdrop, ultimately aiming to proactively address the possible ramifications of the condition, especially in cases of delayed diagnosis.

### 2.1. Assessment of Pathogenic Genetic Variants

Every individual possesses variants within their genome, ranging from single nucleotide substitutions (single nucleotide variants [SNVs] or single nucleotide polymorphisms [SNPs]) [26,27] to duplications or deletions spanning entire chromosomes. On average, each person carries about 100 de novo SNVs that emerge during their development [27]. Given that protein-coding regions make up roughly 1% of the genome, but contain 85% of disease-causing variants, targeted sequencing frequently focuses on these areas [28].

Predominant approaches include sequencing the protein-coding regions of the roughly 20,000 known genes, termed whole-exome sequencing (WES) [29] or focusing on a select set of genes associated with a specific clinical condition, often called a “gene panel” (typically focusing on exons only) [30]. These techniques can identify minor variants, such as SNVs and minor insertions and deletions. However, the identification of larger and more intricate variants, such as the complete deletion of an exon or complex genomic rearrangements, can be more challenging [31,32].

The Sanger sequencing technique has been widely used since 1977 for direct sequencing of specific genes. It also serves to validate mutations detected by alternate sequencing approaches and is pivotal in genetic investigations, particularly when a mutated gene is already identified within a family lineage [29,33]. With the advent of next-generation sequencing (NGS) techniques [34] in the early 2000s [35,36], there has been a shift toward methods that bolster sequencing for increased parallelism and scalability. Presently, high-throughput sequencing stands as the predominant method in many diagnostic contexts [37,38].

NGS techniques are adept at detecting small variants, including SNVs and minor insertions and deletions. These techniques comprise the following:TGP (targeted gene panel): this method examines genes linked to a specific phenotype. However, its scope is primarily confined to selected genes known for their variants, necessitating ongoing updates. The process entails designing a gene panel correlated with a distinct disease and conducting parallel sequencing. It is commonly employed as the primary diagnostic test for probands [31,39].WES: this method is adept at diagnosing probands presenting with diverse disorders, including pediatric and syndromic cardiomyopathies. It encompasses all genes with the objective of sequencing the complete exome. Through an integrated process, data corresponding to the entire exome can be produced, eliminating the need for additional analyses when updated information becomes available [40,41].Whole-genome sequencing (WGS): this technique sequences the complete genome, offering diagnostic insights for probands with varied disorders and detailed data on pharmacokinetic variants. While comprehensive, WGS comes with a greater expense and necessitates intricate data analysis. If panel sequencing yields negative results, both WES and WGS stand as viable subsequent diagnostic options [42].

Table 1 summarizes characteristics, advantages, and disadvantages of each technique.

Alternative quantification approaches that are not based on sequencing include multiplex ligation-dependent probe amplification (MLPA) or array comparative genomic hybridization (aCGH) [32]. These approaches may offer enhanced sensitivity and can complement traditional sequencing methods. Some techniques utilize polymerase chain reaction (PCR) for detecting variants. Allele-specific PCR is an economical and scalable option for specific variant detection. In contrast, digital PCR, encompassing methods such as digital droplet PCR, facilitates precise quantification of target DNA sequence copy numbers relative to a single-copy reference locus. Not only is this technique cost-effective, but it is also adept at detecting subtle dosage variations, making it vital for confirming the existence of potential novel copy number variations (CNVs) identified through sequencing [43]. 

Other genetic investigation techniques involve competitive DNA hybridization using oligonucleotide probes with known sequences. While these approaches might have limited utility in identifying rare variants for Mendelian diagnoses, they are extensively employed in studying common variants. Examples include their application in genome-wide association studies (GWAS), computation of polygenic risk scores (PRSs), and pharmacogenetics [44]. 

#### 2.1.1. Sequencing Modalities

The analytical process begins with the collection of a small sample of peripheral blood or saliva from the patient. At times, samples from parents or close family members may also be required. The DNA essential for sequencing is then extracted from this sample. Upon acquiring the sequences, they undergo analysis and interpretation based on the most current genetic data in the international medical-scientific literature, focusing on the relevant diseases or symptoms. Identified pathological variants are subsequently validated in the original sample using alternative technological methods.

In the context of targeted NGS (targeted gene panel, TGP), the assessment of diseases with pronounced genetic heterogeneity and recognized genes is undertaken using designated panels. These panels are of two main types:Presequencing panels (targeted resequencing): this approach enables the simultaneous analysis of multiple patients by selectively enriching specific genomic regions prior to sequencing.In silico panels (targeted data analysis): this method is applied after exome sequencing and focuses on genes directly associated with the disease under investigation, generally allowing for the analysis of a restricted number of samples in each run [44].

The presequencing panel’s predominant advantage is its cost-effectiveness, attributed to the utilization of desktop sequencers. This method also ensures high coverage in pivotal genomic regions. However, a notable limitation is the consistent requirement to update the panel every time a gene is newly associated with a relevant disease. Conversely, the in-silico panel, while being costlier—given that it typically analyzes only one or two patients simultaneously to attain satisfactory coverage—boasts the benefit of enabling the investigation of newly discovered disease genes in previously sequenced patients. This is achieved without necessitating any alterations to the platform, making it a more economical choice over the long term in comparison to presequencing panels.

Five critical factors influence the selection of the appropriate sequencing method (WES, target sequencing, or WGS): cost, purpose, sensitivity, probability of obtaining incidental findings (IFs) and variants of uncertain significance (VUSs), and data storage [45]. 

-Cost: The difference in cost is influenced by the reagents used. The use of TGP is more economical if the number of samples per run is optimized. In economic terms, WGS is the most expensive, and although WES is pricier than panels, it can be advantageous depending on the type of study to be performed [46].-Purpose: Typically, NGS techniques are employed for diseases with high genetic heterogeneity or Mendelian-based genetic diseases (or those suspected to be genetic) where the causative genes remain unidentified. For diseases with established genetic etiology, either custom-designed panels, WES, or WGS can be utilized [45].-Sensitivity: Sensitivity largely depends on the coverage of the sequences under investigation, that is, the number of reads for specific DNA sections and the overlap extent between these reads. A greater number of reads for a specific region translates to higher sensitivity for that DNA segment [45]. During panel analysis, a reduced genome proportion under investigation leads to enhanced coverage and sensitivity. Thus, if a disease is believed to result from a mosaic genomic alteration, NGS panel analysis offers higher sensitivity compared to WES.-IFs and VUSs: The probability of identifying IFs and VUSs depends on the genome proportion both analyzed and queried. Analysis based on panels has a reduced association with IFs, since the investigated sequences are directly relevant to the clinical presentation of the proband. In general, a broader sequence analysis correlates with a higher number of IFs and VUSs. Notably, VUSs can also appear in targeted analyses of specific gene panels [45].-Data storage: The extent of the genome analyzed directly influences the volume of data generated. Consequently, suitable platforms for data storage are essential, particularly for analyses yielding substantial data, such as those conducted via WES and, more prominently, WGS [45].

#### 2.1.2. Identification and Interpretation of Pathogenic Genetic Variants in Clinical Diagnostics

Various software tools, including Bowtie2, BWA, MAQ, and SOAP2, assist in mapping a patient’s sequences to reference sequences found in databases. Through this method, the patient’s exome is aligned, signifying the alignment phase. After successful alignment, sequence variants can be detected, representing all deviations from the human reference sequence for each gene. This process is known as “variant calling”, an operation that is automated within NGS analysis.

Upon detecting approximately 140,000 genetic variants in an individual (a figure representative of the average number derived from WES) through the previously mentioned variant calling operation, it becomes essential to determine their clinical significance. These variants are characterized based on the following: Their allele frequency within the general population;Their presence or absence in recognized human mutational databases (such as gnomAD, ClinVar [47], or OMIM [48]); andTheir potential influence on protein function as determined by in silico analyses, which produce computational predictions of protein folding using tools such as PolyPhen [49], MutationTaster [50], or PROVEAN [51].

Variants that may have clinical significance are carefully isolated, interpreted, and subsequently reported within a diagnostic framework. The Variant Interpreter software v2.14.0.4 (Illumina) serves as an example tool for this purpose. It facilitates the alignment of sequences to the reference genome (GRCh37/hg19). Additionally, allele frequencies are mapped against the population database gnomAD v2.1.1 (Genome Aggregation Database), all performed automatically by the software.

Guidelines set by the American College of Medical Genetics and Genomics dictate that identified variants should be articulated using specific terminologies such as “pathogenic”, “likely pathogenic”, “uncertain significance”, “likely benign”, and “benign” [52]. In the context of WES/WGS studies, a filtering approach is recommended. For variant interpretation, references are primarily drawn from the established scientific literature, databases such as ClinVar, HGMD, and LOVD, as well as dedicated gene- or disease-specific databases when they are accessible. When considering allele frequencies, the gnomAD v2.1.1 population database and the internal laboratory database are typically referenced. It is crucial to note that the nomenclature and classification of variants, especially those labeled as having “uncertain significance”, might undergo changes influenced by updates in reference sequences and emerging scientific findings. During trio analyses, the genomic data of parents are predominantly utilized to determine the inheritance pattern of variants observed in the proband [52]. Furthermore, secondary variants, which might be indicative of phenotypes not directly related to the primary clinical query, are documented when a comprehensive clinic exome analysis is specifically requested and consented to. Confirmatory tests are performed using secondary DNA extraction to validate the potential clinical cause in a patient.

### 2.2. Variants of Uncertain Significance (VUS)

A mutation denotes a permanent alteration in the nucleotide sequence, whereas a polymorphism represents a variant with a frequency exceeding 1% in the population. Both terms are now more commonly referred to as “variant”, further defined with specific qualifiers such as “pathogenic”, “likely pathogenic”, “uncertain significance”, “likely benign”, or “benign”. Some laboratories may incorporate additional levels, particularly for internal use, such as subclassifying VUSs. The designations “likely pathogenic” and “likely benign” imply a confidence level greater than 90% that the variant is, respectively, pathogenic or benign. Importantly, at present, there is no robust data that provides a quantitative assignment of variant certainty to any of the five categories, largely due to the varied nature of many diseases. Over time, as experimental and statistical methods evolve, they will offer a more objective means of assessing the pathogenicity of variants. The precision of definitions and the desired confidence levels will continue to evolve based on deeper clinical insights and consensus [52].

VUSs are typically overlooked in clinical decision-making, as further information is required to categorize them definitively as either pathogenic or benign. However, the presence of uncertain sarcomere mutations or multiple VUSs in patients with hypertrophic cardiomyopathy (HCM) has been correlated with earlier disease onset and more severe outcomes. Therefore, a deeper understanding of these VUSs becomes imperative for optimized clinical management and better patient outcomes [53]. In an effort to bridge this knowledge gap, experimental tools designed for high-throughput screening of HCM mutations and their pathogenicity have been introduced. These instruments leverage cutting-edge computational methods that harness genetic association networks and polygenic risk prediction models, shedding light on the intricate world of genetic variations linked to diseases. One such tool, VariantClassifier (VarClass), exemplifies this approach [54].

CardioBoost uses a disease-specific variant classifier algorithm tailored to predict the pathogenicity of missense variants linked to inherited cardiomyopathies and arrhythmias. One of its distinct advantages is that, when focused on heart disease, this specific variant classifier surpasses the performance of leading whole-genome tools. This superiority underscores the significant potential for improved pathogenicity predictions through disease-specific approaches [55]. On the other hand, some tools harness detailed structural information about proteins, examining how mutations might affect protein folding and stability to make their pathogenicity predictions [53].

Advancements in technologies leveraging artificial intelligence have enhanced our ability to determine the pathogenicity of mutations in cardiac sarcomere proteins. Given the existing challenges in validating numerous computational algorithms, a methodological approach was undertaken to benchmark variant prediction capabilities, assessing their efficiency in identifying the potential pathogenicity of variants. Through this assessment, certain algorithms stood out for their proficiency in discerning the pathogenicity of HCM variants. Leading tools in this regard include ClinPred, MISTIC, FATHMM, MPC, and MetaLR. Importantly, the combined use of these high-performing tools markedly improves the accuracy of pinpointing the most pertinent VUSs for subsequent analysis [56].

### 2.3. Classification, Implications, and Clinical Utility of Genetic Variants in Cardiovascular Disease

When a genetic variant correlates with an observed phenotype, laboratories typically categorize it using a five-tier system. If strong evidence supports the variant’s causative role in the condition, it will be labeled as either “pathogenic” or “likely pathogenic”. Conversely, if robust evidence suggests that the variant is not the causative agent for the disease, it will be termed “benign” or “likely benign”. In instances where the evidence is either conflicting or not conclusive, the variant is designated as having “uncertain significance”, often abbreviated as “VUS”. It is crucial to note that VUSs should not be used for cascade testing, and rarely hold clinical significance for the proband [57].

Upon identification of a pathogenic or likely pathogenic variant, initiating clinical evaluations and cascade genetic testing for family members is recommended. Family members who test positive for the variant should undergo long-term monitoring, especially if they exhibit the clinical phenotype. In contrast, family members who do not have the variant can generally be exempted from further genetic investigations [22,58]. 

In cases where a VUS is identified, supplementary analyses, such as segregation studies and, where feasible, functional studies, should be conducted. Decisions regarding genetic testing and clinical screenings for family members are based on the findings from these analyses. Equally significant, re-evaluation of other potential causes of the disease is warranted when a genetic variant is absent or a genetic test is inconclusive. Clinical screenings of family members should also be considered. For instances showcasing a typical cardiomyopathy phenotype, long-term monitoring is advised. However, if no other family members are affected, frequent, detailed screenings might be reduced or concluded earlier, while those at higher risk should continue regular screenings [59]. 

PRSs, also known as GRSs, are becoming increasingly relevant in the evaluation of multiple genetic variants [60]. These scores consider variants across the genome, each contributing a small amount to disease risk, to compute a cumulative risk score. However, the definitive clinical utility of PRSs in the management of cardiomyopathies remains to be established [61,62,63,64].

In the contemporary medical landscape, identifying a genetic variant or assessing an individual’s susceptibility to genetic diseases equates to equipping the patient or their family with a genetic profile (genotype). This profile subsequently stratifies risk considering the identified genetic variant, related diseases, coexisting health conditions, and resultant manifestations (phenotype). Such stratifications can indicate events such as SCD or the onset of clinically evident HF. Furthermore, this genetic “identity card” can dictate whether a patient receives a recommendation for an implantable cardioverter-defibrillator (ICD) insertion, along with all the consequential implications it entails [65].

### 2.4. Diagnosis and Counseling

Incorporating genetics into the care regimen for these patients necessitates a structured follow-up and extended surveillance, both for the proband (the initial patient diagnosed with the condition) and their family. This approach not only fosters a more profound comprehension of the disease, but also paves the way for specific and individualized treatments, thereby enhancing both the patient’s prognosis and their clinical management. When diagnosing a patient with cardiomyopathy through genetic testing, the scope extends beyond merely identifying the patient’s present condition. It encompasses continuous genetic assessments for family members and the initiation of a broad counseling framework for both the patient and their family [66,67,68].

For those diagnosed with cardiomyopathy, the genetic trajectory, both for the proband and their relatives, invariably begins with genetic counseling. This step is pivotal, as grappling with an inherited cardiomyopathy can present numerous challenges. Genetic counseling, conducted by healthcare professionals with specific training, strives to assist patients and their families in comprehending and adjusting to the medical, psychosocial, and familial consequences of genetic diseases. It encompasses a range of topics, including understanding inheritance risks, educational sessions on genetic diseases, guidance during genetic testing processes, interpreting genetic variant results, compiling a comprehensive three-generation family medical history, and offering psychosocial support to patients and their families [69]. 

Addressing the communication strategy with family members is a critical component of genetic counseling. The index patient, often referred to as the proband, might choose not to inform at-risk relatives about the inherited condition and the availability of family screening for a variety of reasons [70,71,72]. This situation is concerning, especially given the severe complications and risk of SCD associated with many inherited cardiac conditions. Since many inherited cardiomyopathies exhibit an autosomal dominant inheritance pattern, it is essential to emphasize the importance of both clinical and genetic testing for immediate family members. Barriers to effective communication might encompass strained family dynamics, guilt about potentially transmitting a causative variant to children, associated emotional distress, and difficulty in understanding the implications of results [70,72].

Following the identification of a pathogenic or likely pathogenic variant in an index patient through investigations of relevant disease-associated genes specific to the phenotype, cascade genetic testing can be recommended for first-degree relatives at risk. If a first-degree relative is deceased, evaluating close relatives of the deceased individual should also be taken into account. Continuous genetic screening should be considered for descendants of those relatives confirmed to carry a pathogenic or likely pathogenic variant. Relatives lacking the variant can be discharged from further follow-up, while those with the variant should undergo regular clinical evaluations. Improper use of genetic testing within a family might induce undue anxiety and concerns, potentially leading to challenges due to misunderstandings of genetic interpretations [70].

### 2.5. Gene Therapy

Gene therapy is a prominent strategy within advanced therapies. At its core, it aims to address the root cause of a disease by providing affected cells with a functional version of a gene. Methods of gene editing include RNA therapy, CRISPR/Cas9 techniques, antisense therapies, and the introduction of innovative drugs [73], as summarized in Figure 1. While many clinical phenotypes primarily rely on generic and symptomatic medications, treatments that focus on the underlying cause and offer targeted interventions (specifically, etiology-based and precision therapies) are still in their developmental stages.

Research on inherited cardiomyopathies has encountered obstacles due to the lack of suitable in vitro human cardiac cell or tissue models, especially those that replicate patient-specific anomalies. A promising solution is the generation of human-induced pluripotent stem cell-derived cardiomyocytes (hiPSC-CMs) that are tailored to individual patients [73].

Specialized models of hiPSC-CMs, designed to match specific patients or diseases, have been developed, encompassing mutations in genes encoding cardiac sarcomeric and cytoskeletal proteins, ion channels, nuclear proteins, mitochondrial proteins, and lysosomal proteins. The integration of hiPSC-CM technology with genome-editing methods such as the CRISPR/Cas9 system has yielded deeper insights into the genetic origins of diverse cardiomyopathies, leading to the creation of isogenic control lines [74].

Transitioning from a cellular perspective to three-dimensional engineered heart tissues, made possible by integrating polymer-based scaffolds with hiPSC-CMs, offers more clinically relevant models for cardiomyopathy studies [75,76].

Furthermore, the capacity to modify DNA through genome editing, especially with the CRISPR/Cas9 system, marks substantial progress in cardiovascular medicine. This spans a better comprehension of genetic diseases and the advancement of targeted therapeutic interventions. CRISPR/Cas9, in particular, is instrumental in therapeutic genome engineering for a range of cardiovascular conditions [77,78,79].

For example, gene editing techniques, such as exon skipping, have demonstrated potential in conditions such as Duchenne muscular dystrophy (DMD), in which mutations in the *dystrophin* gene result in the lack of the dystrophin protein [80,81]. Through exon skipping, a truncated but functional protein can be produced. In both animal models and preclinical trials, the application of CRISPR/Cas9 to specifically edit exons in the *dystrophin* gene has shown promise for the future treatment of DMD patients [81].

Additionally, nucleic acid-based drugs, especially oligonucleotides, have opened up new avenues for interfering with disease-related genes, either at the genome or gene expression level. These oligonucleotides have the ability to target various facets of gene expression, including RNA blockade, splice switching, and exon skipping. Various nucleic acid-based drugs have been formulated and approved for different diseases, and their potential applications for cardiomyopathies are under investigation. 

Furthermore, advancements in our understanding of the genetics, epigenetics, and proteomics of cardiomyopathies have paved the way for the development of innovative compounds aimed at previously considered “undruggable” elements. These innovative compounds span a range that includes small molecules, gene-specific therapies, and protein stabilizers. A particular study [82] demonstrated a gene therapy approach that utilized the delivery of *Modulator of Growth 1* (MOG1) gene through adeno-associated virus serotype 9 (AAV9) vectors to upregulate MOG1 itself. This chaperone molecule interacts with protein NaV1.5, facilitating its transportation to the cell surface. This gene therapy method notably mitigated symptoms of cardiac arrhythmia and contractile dysfunction in heterozygous humanized knock-in (KI) mice possessing the SCN5A p.D1275N mutation. The introduction of such a compact chaperone protein presents promising prospects for addressing genes responsible for diseases, especially those pertinent to conditions such as Brugada syndrome, arrhythmias, and certain forms of cardiomyopathy [82].

Table 2 showcases several ongoing clinical trials, either recruiting or already completed, underscoring the continual emergence of novel therapeutic hypotheses in scientific research, with a growing emphasis on targeted therapy strategies (http://clinicaltrials.gov [83], accessed on 15 September 2023). The AAV lacks an envelope and can be tailored to deliver DNA to designated cells. Although various natural serotypes of AAV exist with slight biological differences, they can all be managed using a unified protocol.

Overall, gene therapies for cardiomyopathies appear promising, opening up possibilities for gene-specific treatments, and even treatments tailored to specific pathogenic variants. Despite existing challenges, continuous research and technological innovations consistently advance our understanding and capabilities in cardiovascular genetics and treatment methodologies.

## 3. Genetics, Cardiomyopathies, and Heart Failure

Within the context of cardiovascular diseases, the role of genetics in cardiomyopathies has become increasingly pronounced. Cardiomyopathies represent a varied group of heart muscle conditions that manifest with different levels of systolic and/or diastolic cardiac dysfunction. The etiology of many cardiomyopathies is rooted in genetics, frequently resulting from mutations in genes that encode ion channels, sarcomeres, or cytoskeletal elements alongside established acquired causes. The surge in cardiomyopathy research can be attributed to cutting-edge investigative techniques such as induced pluripotent stem cells, 3D cell printing, support structures, and the creation of engineered cardiac tissues [73].

Cardiomyopathy is characterized as a myocardial disorder in which the heart muscle exhibits structural and functional abnormalities occurring distinctly from factors such as coronary artery disease, hypertension, valvular disease, or congenital heart disease that could provide alternative explanations for the myocardial abnormalities observed [95]. Cardiomyopathies are responsible for approximately one-third of clinically observed HF cases. Recent guidelines from the European Society of Cardiology (ESC) reaffirm conventional cardiomyopathy classifications such as hypertrophic cardiomyopathy (HCM), dilated cardiomyopathy (DCM), restrictive cardiomyopathy (RCM), and arrhythmogenic right ventricular cardiomyopathy (ARVC). Additionally, there is a proposition to introduce the term “non-dilated left ventricular cardiomyopathy” (NDLVC). This would cater to those previously labeled under various terms such as DCM without left-ventricular dilation, arrhythmogenic DCM (that does not fit the ARVC criteria), arrhythmogenic left ventricular cardiomyopathy (ALVC), and left-dominant ARVC.

HCM is prevalent in approximately 0.2% to 1.4% of the general population [96,97]. Up to 60% of HCM cases can be traced back to pathogenic genetic variants, predominantly stemming from mutations in sarcomeric genes, typically inherited in an autosomal dominant manner. While most cases are genetic, approximately 10% can be attributed to other causes, such as neuromuscular disorders, storage diseases, and certain hereditary syndromes. Among those with HCM, 40–70% present with obstructive HCM (manifested either at rest or during exercise), characterized by a marked reduction in ventricular cavity size. Conversely, 30–60% exhibit nonobstructive HCM. The primary genes implicated in HCM are *myosin heavy chain 7* (MYH7) and *myosin-binding protein C3* (MYBPC3) [98,99].

DCM’s prevalence in Europe is estimated to fall between 0.036% and 0.4% [100,101]. 

In this demographic, only 30–40% of cases are linked to rare pathogenic genetic variants, suggesting a substantial role of polygenic or common variant influences. Disease acceleration can be attributed to modifiers such as epigenetic factors or acquired elements such as pregnancy, hypertension, alcohol abuse, and toxin exposure. Although genetic variants are more prevalent in families impacted by DCM, about 20% of these variants are also discernible in nonfamilial cases [100].

The prevalence of NDLVC is yet to be precisely established. Genes associated with this phenotype include *desmin* (DES), *lamin A/C* (LMNA), *phospholamban* (PLN), *filamin C* (FLNC) with truncating variants, and *desmoplakin* (DSP) variants, which are frequently linked to left ventricular fibrosis. Recognizing a pathogenic or likely pathogenic genetic variant enhances the accuracy of prognosis and outcome predictions [102].

ARVC is an inherited condition associated with genes encoding desmosomal proteins, with an estimated prevalence of approximately 0.078% [103,104]. The genes implicated in ARVC are responsible for the proteins of the cardiac desmosome, such as *plakophilin-2* (PKP2), DSP, *plakoglobin* (JUP), and *desmoglein-2* (DSG2). Other genes, including *transmembrane protein 43* (TMEM43) [105], PLN [106], and DES [107], have been identified as having pathogenic or likely pathogenic variants. 

RCM is relatively rare among adults. It typically exhibits autosomal dominant inheritance, though autosomal recessive or sporadic cases are also observed. RCM is linked to mutations in genes for sarcomeric structural and regulatory proteins, as well as cytoskeletal filaments. The most frequently implicated gene is *troponin I* (TNNI3) [108] or, less frequently, *troponin T type 2* (TNNT2), *titin* (TTN), and *myopalladin* (MYPN).

### 3.1. Genetic Factors Predisposing to Heart Failure

In patients with cardiomyopathy and those with a genetic variant (i.e., carrier), it is crucial to identify those at an elevated risk of developing HF. The proportion of patients possessing a genetic variant linked to cardiomyopathies who eventually develop HF can differ substantially. This variability can be attributed to multiple factors, including the specific genetic variant in question, its severity, the patient’s family history, age, and other individual risk determinants [100]. At present, there are no well-established and scientifically validated multiparametric risk scores that can precisely predict which patients will manifest HF. Nevertheless, a comprehensive evaluation of individual risk should incorporate several parameters. In this context, family history and genetic information can be instrumental in identifying heightened risk. It is well-established that having a first-degree relative with HF augments the risk relative to the broader population [100,109]. A cross-sectional cohort study, which encompassed 458 participants, each with at least one parent diagnosed with HF, revealed that asymptomatic participants exhibited a higher likelihood of increased left ventricular systolic dysfunction, left ventricular internal dimensions, and left ventricular mass [109]. The study did not explicitly indicate whether patients with HF possessed specific genetic variants. However, a clear hypothesis emerged, suggesting that the inheritance of certain genetic factors (those linked to maladaptive reactions to environmental or biological stressors) implies a causal relationship between these genetic elements and the disease’s progression [109]. 

This familial predisposition could potentially stem from shared genetic variants. Various genetic variants have been associated with different types of cardiomyopathies, which can significantly influence the course and outcome of the disease. For example, in DCM, there is evidence that patients with TTN truncation variants might be more susceptible to developing HF when exposed to stressors such as alcohol, pregnancy, or cardiotoxic drugs. Furthermore, those with DCM who carry sarcomeric rare variants or *RNA-binding motif protein 20* (RBM20) variants generally experience a more rapid disease progression, sometimes leading to the necessity for interventions such as heart transplantation. Variations in the LMNA gene can have diverse effects on cardiomyocyte structure and function. Notably, non-missense LMNA variants have been identified as predictors of severe ventricular arrhythmias (VAs) and have been integrated into risk assessment models [110]. 

In the context of HCM, a large multicenter cohort study has shown that a sarcomere mutation is associated with an earlier onset of the disease and serves as a strong predictor of unfavorable clinical outcomes, including the occurrence of VAs and HF [111]. The MYH7 gene is a significant genetic factor responsible for encoding the *β-myosin heavy-chain* (β-MHC) subunit of cardiac myosin. Alterations in this myosin directly influence myocardial mechanical function, contributing to impaired myocardial performance that can result in HF [112].

Individuals carrying MYBPC3 missense VUSs demonstrate an elevated incidence of adverse clinical outcomes. These outcomes include VAs, HF, all-cause mortality, atrial fibrillation, and stroke, similar to those with pathogenic MYBPC3 variants [113]. Evidence indicates that patients with MYH7 gene mutations present with more severe disease manifestations compared to those with MYBPC3 mutations [114]. Moreover, patients harboring MYH7 or MYBPC3 mutations also face a heightened risk of SCD [115]. 

In the context of ARVC, it has been observed that patients with mutations in the DSP, *desmocollin-2* (DSC2), and DSG2 genes more commonly exhibit a diminished left ventricular ejection fraction (LVEF) of ≤45% on cardiac magnetic resonance (CMR) imaging in comparison to those with PKP2-related ARVC (27% vs. 4%, *p* < 0.01) [116]. 

A recent meta-analysis revealed that mapping of TNN variants identified specific regions within the TNNT2 and TNNI3 genes associated with increased pathogenicity in RCM and a heightened risk of SCD. TNNC1-positive probands presented with the earliest age of onset and the highest incidences of death, transplantation, or ventricular fibrillation events. Notably, these individuals were diagnosed at younger ages and experienced adverse clinical outcomes, with some facing fatal outcomes during infancy [117]. 

Patients with genetic variants predisposing them to an elevated risk of HF and SCD warrant vigilant monitoring to facilitate early diagnosis [118]. It is pertinent to highlight that morpho-functional alterations can manifest long before the emergence of congestive symptoms. Furthermore, the reported absence of symptoms often mirrors a gradual adaptation to the constraints imposed by the cardiac condition, a process that might begin during adolescence [119]. 

Monitoring should leverage a multimodal imaging approach, integrating both transthoracic echocardiography and CMR. Transthoracic echocardiography yields invaluable diagnostic information for cardiomyopathies, and aids in identifying patients at an elevated risk of HF. In conditions such as DCM, a decrease in LVEF is a vital indicator of HF severity, signifying compromised ventricular contractility and concomitant ventricular wall thinning [118,120]. 

Advanced myocardial deformation imaging techniques, such as speckle tracking and tissue Doppler, are particularly adept at assessing global longitudinal strain. These methods prove to be more discerning than EF in detecting nuanced ventricular dysfunction and are ideal for preliminary screenings. In contrast, CMR offers superior tissue characterization. It is indispensable for diagnosing conditions such as NDLVC, ARVC, myocarditis, amyloidosis, and sarcoidosis. Furthermore, these imaging techniques are essential for gauging the progression of diseases and the success of therapeutic approaches. The pattern and extent of late gadolinium enhancement (LGE) provide valuable insights into the prognosis for arrhythmia and the severity of HF. Periodic monitoring through echocardiography or magnetic resonance imaging can uncover abnormalities in cardiac dimensions or operations ahead of apparent symptom manifestation, thereby identifying a subclinical phenotype [120]. Additionally, laboratory assessments, including natriuretic peptides and troponin levels, are beneficial for overseeing cardiomyopathies and pinpointing HF [121]. 

### 3.2. Genetics in Assessing the Risk of Heart Failure Progression

Genetics can be pivotal in guiding the clinical management of patients diagnosed with cardiomyopathy, especially those with established HF. Understanding both the patient’s clinical phenotype and genetic background provides the HF specialist with a unique profile (genotype) for that specific patient, enabling more accurate predictions regarding disease progression and outcomes.

Currently, there are no predefined and scientifically validated multiparametric risk scores that can accurately project which patients will develop HF or experience clinical deterioration [100,109]. Nonetheless, certain genetic variants have been linked to a heightened risk of malignant VAs and progression to advanced HF [4,110].

Routine clinical evaluations, complemented by echocardiography or CMR every 1–2 years and cardiopulmonary exercise testing every 2–3 years, serve as reliable tools to detect early indicators of disease progression. In this context, timely clinical evaluation and preventive strategies are of paramount importance. Leveraging genetic insights allows for enhanced risk stratification in patients with cardiomyopathy [122].

### 3.3. Arrhythmic Risk Stratification

In individuals with cardiomyopathies, atrial fibrillation stands out as the most prevalent arrhythmia across all phenotypes. This condition is linked to an elevated risk of HF, cardioembolic events, and increased mortality. The primary concern for clinicians, however, is managing the risk of VAs leading to SCD, as well as sustained VAs and electrical storms [123]. Indeed, while the implantation of an ICD is advocated for secondary prevention, determining its use in patients who have not yet experienced sustained symptomatic VAs remains a complex clinical decision.

In DCM, individuals carrying pathogenic variants such as DSP, LMNA, PLN, *folliculin* (FLCN), TMEM43, and RBM20 demonstrate an increased incidence of notable arrhythmic events [124,125]. This elevated risk is observed irrespective of their LVEF, and these patients also tend to follow a more challenging clinical trajectory compared to DCM patients not carrying these specific genetic variants. Genes associated with an increased risk of arrhythmias include those encoding nuclear envelope proteins (TMEM43, LMNA, and *emerin* [EMD]), as well as cytoskeletal and desmosomal proteins (PKP2, DSP, DSC2, and DSG2) [122].

Patients with DCM-causing variants in these high-risk genes should be viewed as having a pronounced genetic predisposition to SCD. The decision to implant an ICD for primary prevention in those with LMNA-related cardiomyopathies might be informed by the LMNA-risk ventricular tachyarrhythmia (VTA) calculator, which evaluates the risk of experiencing a life-threatening VTA within a 5-year period [122]. This calculator provides a more nuanced risk assessment compared to the prevailing standard of care [4], which identifies high risk based on the presence of ≥2 of the following factors: male gender, non-missense mutations, nonsustained ventricular tachycardia, and an LVEF of ≤45%. Notably, LMNA mutations have been linked to premature atrial and ventricular arrhythmias, early-onset conduction abnormalities, an elevated risk of SCD, and progression to end-stage HF. Consequently, the LMNA-risk VTA calculator was developed specifically for this patient subset, aiming to quantify the likelihood of SCD, relevant ICD shocks, or manifestations of hemodynamically unstable VTA (https://lmna-risk-vta.fr [126], accessed on 15 September 2023). 

In individuals without a high-risk genotype and an LVEF ≥ 35%, the use of LGE on CMR imaging becomes an essential tool for risk assessment [127]. It is recommended that genetic testing, encompassing at least the PLN, LMNA, FLNC, and RBM20 genes, be conducted for patients with DCM and atrioventricular (AV) conduction delay before the age of 50, or if a family history of SCD in a first-degree relative is present [122]. For DCM patients carrying a pathogenic LMNA gene mutation, considering the placement of an ICD for primary prevention is essential, particularly if the projected 5-year risk of encountering life-threatening VA is ≥10%. This consideration is heightened when the coexistence of nonsustained ventricular tachycardia (NSVT) with an LVEF of ≤50% or an AV conduction delay is observed [122,128]. Additional determinants, such as syncope, the presence of NSVT, or a significant incidence of ventricular ectopy (VE), can further influence the decision regarding ICD implantation [122,125].

MYH7-related dilated cardiomyopathy (MYH7-DCM) is characterized by an early onset, pronounced phenotypic manifestation, limited left ventricular reverse remodeling, and a frequent progression to end-stage HF (ESHF). Complications related to HF are more common than the occasional VAs [129].

In HCM, NSVT identified during extended (24/48 h) ambulatory electrocardiogram (ECG) monitoring, particularly in individuals below 30 years of age and those who experience NSVT during physical exertion, is significantly associated with an increased risk of SCD [130]. A risk-prediction model, known as the HCM Risk-SCD, takes into account parameters such as left ventricular wall thickness, the gradient of the left ventricular outflow tract (LVOT), left atrial dimensions, unexplained episodes of fainting (syncope), NSVT, age, and a family history of SCD [131]. However, the HCM Risk-SCD model does not consider certain elements, including the presence of an apical aneurysm, compromised left ventricular systolic function, and pronounced LGE on CMR.

In addition to these tools, the presence of single or multiple sarcomeric mutations also plays a significant role in the stratification of SCD risk [132]. 

In the context of ARVC, genetic testing reveals mutations in 4–16% of cases [133]. These identified mutations are associated with a heightened propensity for VAs to manifest at an earlier age. Identifying ARVC patients at heightened risk for SCD is challenging, due to the scarcity of comprehensive evidence outlining definitive risk factors for life-threatening VAs. Beyond genetic variants, clinical presentations, such as arrhythmic syncope and both right ventricular and left ventricular dysfunction, have been recognized as significant contributors to an increased arrhythmic risk [122]. 

### 3.4. Implications of Genetic Variants in Clinical Decisions

As previously mentioned, determining the genetic variants within a patient through genetic testing provides a distinctive “identity card”, revealing potential clinical phenotypic traits of the individual. Such characteristics might predispose them to malignant VAs, potential SCD, and the initiation and progression of HF symptoms.

Table 3 showcases a gene panel related to specific cardiomyopathies. It differentiates genes that predominantly manifest with an arrhythmic risk profile, those leaning toward an HF clinical outcome, and mutations that predispose individuals to both conditions [108,110,113,116,129,134,135,136]. This underscores the importance of genetic investigations in predicting prognosis, accurately classifying the patient’s genotype, initiating early medical surveillance, and preemptively addressing potential adverse cardiovascular events.

Individuals harboring mutations in genes such as DSP, LMNA, PLN, FLCN, RBM20, and TMEM43 are more susceptible to significant arrhythmic events, irrespective of their LVEF, and tend to follow a more challenging clinical trajectory than those with DCM who are genotype-negative. Specifically, a patient with a mutation in the LMNA gene faces an elevated risk of SCD. As a result, the criteria for ICD implantation for these individuals will be more stringent compared to those without such a mutation, but who are still at high risk for SCD. Consequently, patients in this category should consider ICD implantation as a precaution, even when their LVEF is above 35% [4,122].

Recognizing one’s genetic profile and inherent predispositions will gain paramount importance in the future, as numerous scientific studies have already highlighted. For instance, in peripartum cardiomyopathy, which was historically viewed as an environmental condition, there is a notable prevalence of truncating variants in genes associated with DCM. Such variants correlate with a diminished rate of left ventricular recovery. By comprehending these genetic nuances, clinicians can stratify patients with peripartum cardiomyopathy and tailor therapeutic approaches accordingly [137]. 

Thus, it becomes essential to embrace the idea of individual genetic predispositions. This suggests that specific genetic variants might have a direct correlation with either favorable or unfavorable outcomes. This is even applicable to less prevalent cardiomyopathies, where previously only external or secondary causes were considered as potential etiologies [138]. 

## 4. Heart Failure and Genetics: Beyond Cardiomyopathies

HF emerges as a multifaceted outcome shaped by the interplay of genetic, acquired, and environmental dimensions. Common genetic variants can precipitate its onset, suggesting that these variants, in conjunction with environmental factors, amplify the susceptibility to HF. Empirical evidence robustly underscores the hereditary dimensions of HF. The Framingham database [139,140], for instance, revealed a marked 70% increase in the likelihood of HF development in individuals with a family history of the condition. Nonetheless, it is pivotal to recognize that numerous cases are also influenced by lifestyle and external environmental determinants. The trajectory of HF evolution is determined by individual predispositions coupled with the intrinsic factors of accompanying comorbidities. Genetic predispositions can influence susceptibility to the initial event that triggers HF, dictate the progression post-onset, and modulate the efficacy of therapeutic strategies [141].

Figure 2 summarizes the complexity of the influence of genetic background observed in HF patients. 

Differentiating between acquired and inherited forms of HF presents a nuanced challenge. We advocate for a comprehensive exploration to uncover the potential genetic underpinnings of the condition. This approach assists in identifying family members at heightened risk for HF well before the manifestation of clinical symptoms [122]. Additionally, strategic counseling for patients might encompass recommendations such as avoiding competitive sports, initiating cardioprotective treatments to modulate the disease’s progression, mitigating cardiac decompensation or remodeling phases, and taking proactive measures to minimize the risk of SCD [142].

## 5. Current Guidelines and Recommendations

The recent ESC guidelines dedicated to cardiomyopathies provide comprehensive recommendations regarding genetic testing. For individuals with cardiomyopathy, undergoing genetic testing, often for confirmatory or diagnostic purposes, offers multiple benefits: it aids in affirming the diagnosis, providing prognostic insights, guiding therapeutic choices, and assisting reproductive decisions [122]. While genetic testing might not significantly alter the management approach for patients suffering from HF, especially in advanced stages, its indication remains crucial if it could benefit family members. This consideration gains prominence for relatives under prolonged monitoring, especially when the genetic cause remains unidentified. However, such concerns can be mitigated if a family-wide genetic diagnosis is ascertained [143].

Upon detection of a pathogenic or likely pathogenic variant in an index patient after analyzing pertinent disease genes aligned with their specific phenotype, cascade genetic testing for first-degree relatives at potential risk becomes feasible, accompanied by preliminary genetic counseling. In cases where a first-degree relative has passed away, consideration should also be given to assessing the immediate relatives of the deceased individual (i.e., the second-degree relatives of the index patient) [122]. 

Genetic counseling, provided by trained healthcare professionals [8], is recommended for families either affected by or suspected to have an inherited cardiomyopathy, regardless of whether they opt for genetic testing. The process of genetic testing for cardiomyopathies should occur under the guidance of a multidisciplinary team proficient in genetic testing methodologies, the interpretation of sequence variants, and the clinical implications of such tests [122]. Typically, this approach is most effectively managed within specialized cardiomyopathy services, or within a network model with equivalent expertise. It is essential that individuals undergoing genetic testing for cardiomyopathy receive both pre- and post-test counseling [144,145].

For the index patient, genetic testing is recommended when it aids in diagnosis, prognosis, therapeutic planning, reproductive management, or the cascade genetic assessment of their relatives [146,147].

Genetic testing is proposed for deceased individuals diagnosed with cardiomyopathy postmortem, especially if it benefits the management of surviving family members. In situations where the testing does not directly aid in diagnosis, prognosis, therapeutic strategy, or cascade genetic analysis of relatives, such testing can still be contemplated if it yields an overall benefit to the patient [148]. There are multiple potential outcomes when conducting genetic testing on the proband. A comprehensive decisional algorithm and workflow are detailed in the ESC Guidelines [8].

Both the ESC guidelines concerning cardiomyopathy and HF emphasize the relevance of genetic insights in deciding upon ICD implantation. As highlighted previously, if certain genetic variants correlate with a heightened risk of SCD, ICD consideration might extend to DCM patients with an LVEF > 35% [122]. 

The guidelines from the American Heart Association (AHA), American College of Cardiology (ACC), and Heart Rhythm Society (HSSA) for managing HF, released in 2022, recommend that for patients with a suspected genetic or hereditary cardiomyopathy, a thorough analysis of the family history, encompassing a minimum of three generations, should be conducted, and a family pedigree should be established [7]. The purpose of these analyses is to identify patients for whom genetic testing could yield significant insights. For certain patients diagnosed with genetic or hereditary cardiomyopathies, genetic screening of their first-degree relatives is advised. The primary goal of this screening is the early detection of cardiac conditions, facilitating swift interventions that could curb the progression of HF and reduce the risk of SCD [7]. 

Currently, no medical treatments tailored to an individual’s genetic makeup or gene therapy are endorsed.

## 6. Genetics and Heart Failure in Clinical Practice

In patients affected by HF caused by cardiomyopathy, the presence of pathogenic genetic variants holds significant implications for clinical management. Successful genetic testing invariably involves at least three key professionals: the geneticist, the cardiomyopathy specialist, and the HF specialist. Their collective expertise becomes invaluable in both patient and family management, as illustrated in Figure 3. The geneticist’s role, detailed throughout this review, is central to identifying pathogenic variants and offering appropriate counseling to patients and their families [8]. Over recent years, the collaboration between cardiomyopathy and HF specialists has gained momentum, proving instrumental in defining genotype–phenotype correlations, evaluating VUS [45] cosegregation, and conducting family screenings. The involvement of the HF specialist, in particular, carries multifaceted significance. 

Primarily, specialists in HF must comprehend the clinical implications of specific genetic variants in a patient’s medical trajectory. Moreover, the identification of a genetic variant is paramount for family members when a patient with cardiomyopathy progresses to HF. Collaboration with the geneticist gains further significance in the context of potential advancements in gene therapy. The HF specialist plays a critical role in educating patients and their families, facilitating a comprehensive understanding of the disease, its associated risks, and available treatments. Additionally, the HF specialist can contribute to research endeavors aimed at discerning the association between genetics and cardiomyopathies, contribute to genetic data collection, deepen insights into these conditions, and ensure individualized patient care.

Furthermore, certain inherited cardiovascular diseases might manifest with multiorgan involvement. This underscores the importance of a collaborative framework where diverse medical professionals—from geneticists and HF specialists to cardiomyopathy experts—collaborate to devise optimal therapeutic strategies for affected individuals.

## 7. Conclusions

In accordance with the recent ESC guidelines on the management of cardiomyopathies, considering the clinical trajectory of this condition and its notable prevalence, especially as a proportion of HF cases, elucidating the genetic background of individuals affected by cardiomyopathies has gained paramount importance [122]. A collaborative approach involving HF specialists, geneticists, and echocardiographers is crucial for a comprehensive clinical evaluation and informed prognosis of these patients. While the future seems promising, the current landscape lacks formally approved and definitive gene therapies for this disease. The precision of patient evaluations and prognostications will likely be enhanced with rigorous data collection, progressive scientific research, and the expansion of known pathological variants and VUSs. Given the nuances of variable expressivity and the challenge of incomplete penetrance, it is vital to incorporate these factors into the overall management strategy for cardiomyopathies. Such an integrated approach not only provides advantages to patients and their families, but also aids in the early detection of potential complications.

## Figures and Tables

**Figure 1 ijms-24-15221-f001:**
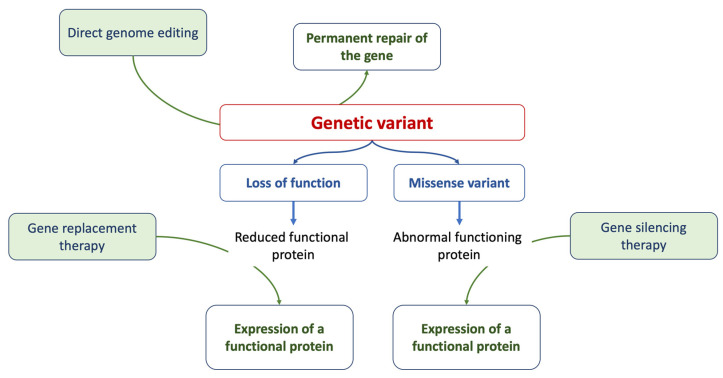
The figure summarizes the current approaches of gene therapy.

**Figure 2 ijms-24-15221-f002:**
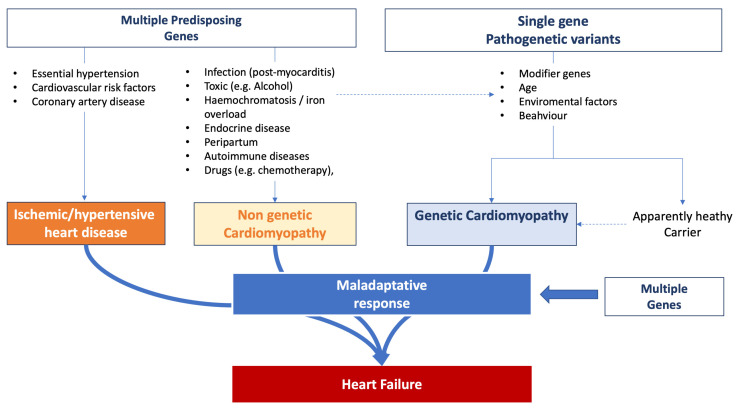
Influence of genetic background on heart failure onset and progression.

**Figure 3 ijms-24-15221-f003:**
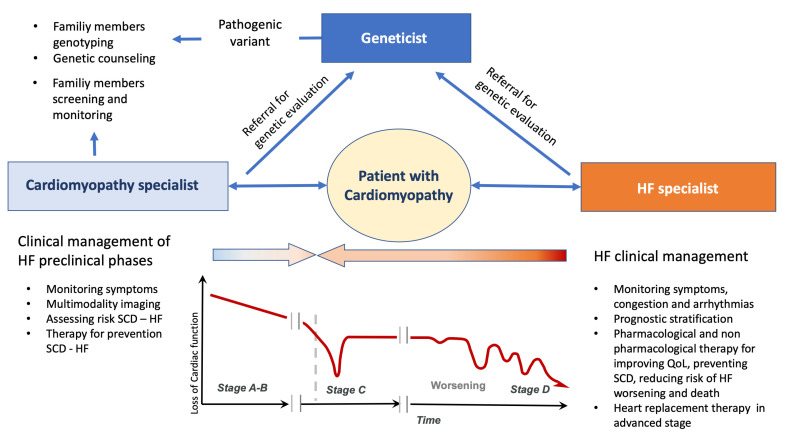
The management of inherited cardiomyopathy should be based on a multidisciplinary approach in which the specialists for heart failure and cardiomyopathies cooperate with the geneticist, in the different stage of heart failure, in order to better manage the patients and their family members. HF: heart failure; QoL: quality of life; SCD: sudden cardiac death; VAD: ventricular assist device.

**Table 1 ijms-24-15221-t001:** Characteristics, advantages, and disadvantages of the most used techniques for detection of genetic variants.

Technique	Characteristic	Applications	Advantage	Disadvantages
WES	Sequencing of the entire exonic genome portion	Mendelian diseases with unknown genes Multifactorial diseases Identification of such susceptible genes	Can be re-evaluated in the future, even in the case of a negative result Less expensive than WGS	Not very sensitive Interpretation of VUS
WGS	Sequencing of the entire genome, both exonic and intronic	Mendelian diseases with unknown genes Multifactorial diseases Identification of such susceptible genes Gene regulation studies	Can be re-evaluated in the future, even in the case of a negative result	Not very sensitive Need for adequate data collection platforms The most expensive
TGP	Panel of known genes that can be sequenced simultaneously Two types:	Mendelian diseases with known genes Polygenic diseases	The most sensitive Customizable Lowest cost	Interpretation of VUS Need to update the panel Higher costs than WES/WGS
	Targeted Data Analysis Panel constructed with genes known to be associated with a specific disease following a WES		Extend the analysis to the rest of the exome in a research context	Need to carry out a preliminary WES
	Targeted ResequencingEnrichment of specific genomic regions before sequencing		Analysis of multiple patients at the same time Use of benchtop sequencers Less expensive than targeted data analysis	Continuous update of new genes associated with a pathology

TGP: targeted gene panel; WES: whole exome sequencing; WGS: whole genome sequencing.

**Table 2 ijms-24-15221-t002:** Ongoing trials evaluating the effects of gene therapy in cardiomyopathies.

Technique	Setting	Treatment	Gene	Phase Study	Ongoing Study ID
AAV	Friedreich’s ataxia DCM	AAVrh.10hFXN	*Frataxin* (FXN)	I	NCT05302271 [84]
AAV serotype 1	LGMD2D	rAAV1.tMCK.hαSG	*Alpha-sarcoglycan* (haSG)	I	NCT00494195 [85]
AAV	ARVC	RP-A601	*Plakophilin-2a* (PKP2a)	I	NCT05885412 [86]
AAV serotype 1	Limb girdle muscular dystrophy type 2C	Gamma-sarcoglycan vector injection	SGCG	I	NCT01344798 [87]
AAV	Adv-HF	MYDICAR	SERCA2a	I/II	NCT02346422 [88]
AAV serotype 1	CHF	AAV1-CMV-SERCA2a	SERCA2a	II	NCT01966887 [89]
AAV	Friedreich’s ataxia with evidence of cardiomyopathy	LX2006 (AAVrh.10hFXN)	hFXN	I/II	NCT05445323 [90]
Drug PF-07265803	REALM-DCM	ARRY-371797	*Lamin A/C*	III	NCT03439514 [91]
AAV	MYBPC3 nHCM	TN-201	MYBPC3	I	NCT05836259 [92]
Genetic (VEGF1)	CHF	VEGF1	phVEGF165	I	NCT00279539 [93]
AAV serotype 9	Danon Disease	RP-A501	LAMP2B	I	NCT03882437 [94]

AAV: adeno-associated virus gene transfer vector; Adv-HF: advanced heart failure; CHF: chronic heart failure; LGMD2D: limb girdle muscular dystrophy type 2D; REALM-DCM: dilated cardiomyopathy with *lamin A/C* gene mutation; SGCG: *gamma-sarcoglycan protein.*

**Table 3 ijms-24-15221-t003:** Clinical relevance of mutations for risk of arrhythmic events and heart failure progression in cardiomyopathies.

Cardiomyopathy	Gene (Protein)	Arrhythmic Risk	Heart Failure Risk
DCM	LMNA (*Lamin A/C*)	X [110]	
	TTN (*Titin*) truncating mutations		X [106,134]
	RBM20 (*RNA-binding motif protein 20*)	X [134]	X [134]
	PLN (*Phospholamban*)	X [135]	
	FLNC (*Filamin C*)	X [135]	
	TMEM43 (*Transmembrane protein 43*)	X [106]	
	EMD (*Emerin*)	X [136]	
	DSG2 (*Desmoglein 2*)	X [106]	
	DSP (*Desmoplakin*)	X [106]	
	DES (*Desmin*)	X [106]	
	MYH7 (*Beta-myosin heavy chain*)		X [129]
HCM	MYBPC3 (*Cardiac myosin binding protein C*)	X [106]	X [113]
	MYH7 (*Beta-myosin heavy chain*)	X [106]	X [129]
RCM	TNNT2 (*Troponin T2*)	X [106]	
	TNNI3 (*Troponin T3*)	X [108]	
ARVC	DSC2 (*Desmocollin 2*)		X [116]
	DSG2 (*Desmoglein 2*)		X [116]
	DSP (*Desmoplakin*)		X [116]

DCM: dilated cardiomyopathy; HCM: hypertrophic cardiomyopathy; RCM: restrictive cardiomyopathy; ARVC: arrhythmogenic cardiomyopathy.

## Data Availability

Not applicable.

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
