# Peer review of "The Role of Genetics in the Management of Heart Failure Patients"

_ijms, 2023, doi:10.3390/ijms242015221_

Round 1
Reviewer 1 Report
Dear Authors,
I would like to express my appreciation for your comprehensive and detailed work. Your paper effectively navigates the journey from identifying a genetic variant to making healthcare decisions that enhance the quality of life for patients and their families. It provides valuable insights into the factors influencing decision-making, the impact of gene variants on cardiac function, and the associated risks and prognoses for individual patients. While I commend your work, I also believe that there is room for improvement through some necessary adjustments and suggestions.
One primary concern in the manuscript is the inconsistent use of acronyms. Some acronyms are defined multiple times in the paper, while others are defined unnecessarily. Additionally, certain paragraphs are complex and require greater clarity or better alignment of ideas. Lastly, I observed a lack of citations in critical sections that demand proper sourcing of information.
In the following paragraphs, I will provide a detailed list of the required adjustments:
- Line 37: "heart failure" should be initially defined as HF. Afterward, you can use the acronym consistently throughout the paper.
- Lines 25 to 29: Citations are required for the information presented in these lines.
- Line 88: The term "the proband" is introduced for the first time. Please clarify the meaning of this term in the context of genetic testing.
- Line 101: The acronym "WES" is not defined, and it should be introduced upon its first use.
- Lines 106: The meaning of "Variant’s boundary" (the breakpoint) is not clearly defined.
- Lines 104 to 129: These lines discuss sequencing techniques. I recommend creating a concise figure summarizing the characteristics, advantages, and disadvantages of each technique for clarity.
- Lines 123 to 124: The phrase, "It can perform a single wet-lab workflow and update the analysis to incorporate new knowledge without regenerating data [40,41]," lacks clarity and doesn't contribute significantly to the message. Please clarify its meaning.
- Line 139: The term "survey method" needs clarification.
- Lines 201 to 208: This paragraph appears complex and would benefit from a clearer message. I recommend rephrasing it. Additionally, the phrase "to name a few" has an informal tone and should be omitted.
- Line 228: It is advisable to define the acronym "VUS" in the title since it is used multiple times in this section.
- Line 273 to 274: The phrase, "VUSs should not be used for cascade testing and are rarely actionable in the proband [57]," lacks clarity. Please rephrase it for better understanding.
- Line 298: Please define "ICD" upon its first use.
- Line 302: Although "proband" is defined here, it should have been defined in its first appearance (Line 88).
- Line 209: The term "proband" is defined again, which could be redundant.
- Line 321: The acronym "ICC" is defined here, but it is not used elsewhere in the manuscript. Therefore, I recommend reviewing the entire manuscript for acronym definitions and using them only when the concept is mentioned more than twice.
- Line 322: The acronym "SCD" is defined here, even though it was mentioned earlier (Line 66).
- Figure 1: Consider replacing the left box, "prevention of abnormal protein expression," with "expression of a functional protein" or a similar phrase for clarity.
- Lines 371 to 376: Citations are required for the information presented in these lines.
- Line 382: Define "AAV9," but also explain the relevance of using the AAV technique and the significance of different serotypes. This clarification would be particularly helpful in understanding Table 1, which lists several AAV serotypes.
- Lines 429 to 436: The citation [10] does not appear to reference DCM, and there are no references for the rest of this paragraph. Please review the citations throughout the article.
- Line 438: There is a typographical error; "PLN gene" appears as "PNL." Lines 452 to 453: The phrase, "those with a genetic variant (genotype-positive, phenotype-negative)," is somewhat complex and can be clarified by equating it to the concept of a "carrier."
- Lines 455 to 466: Citations are needed here. Additionally, "heart failure" should be referred to as HF.
- Line 463 to 464: It is unclear whether the parents mentioned in the sentence have a genetic variant. Please provide clarification.
- Line 468: The term "Genetic Variant" is capitalized, which seems inconsistent.
- Line 492: There is a duplication of the definition of "ARVC."
- Line 498: There is a duplication of the definition of "RCM."
- Line 498 to 499: The phrase, "TNNC1-positive probands with mutations in the TNNC1 gene," appears redundant.
- Line 510: Please define "CMR" upon its first use.
- Line 522: Use "CMR" when referring to magnetic resonance imaging (MRI).
- Lines 526 to 537: Citations are required for the information presented in these lines.
- Line 547: You have defined "SCD" again.
- Line 549: You have defined "ICD" again.
- Line 566: The acronym "LGE" is not defined. Consider whether it is necessary to use the acronym here.
- Lines 571 to 572: Please define "VA" and "AV," if necessary, as they are not commonly used across the manuscript.
- Table 2: Clarify how the risk classification was defined. Incorporate citations in the table to provide the basis for this classification.
- Lines 616 to 619: Consider using "SCD" for consistency. The text in this section is complex; aim for simplicity and clarity.
- Line 622: The definition of "PPCM" appears unnecessary since it is mentioned only once again.
- Lines 627 to 630: This paragraph lacks clarity. Please rephrase it to enhance understanding.
- Line 642: Suggest removing the word "tries."
- Figure 2: The color of the box labeled "Maladaptive response" may not provide sufficient contrast for the arrows behind it. Please use a color with a higher contrast.
- Line 651: Use "SCD" for consistency.
- Line 698: You have defined "SCD" again.
-
Line 712: The acronym "VUS" is defined again.
-
Line 713: I recommend using "HF" consistently after defining "heart failure" in its first appearance.
-
Lines 718, 723, 726, 737: Please use "HF" for "heart failure" consistently throughout the text after its initial definition.
-
Line 744: The acronym "VUS" is defined again.
There are some complex paragraphs that need greater clarity or better alignment of ideas
Author Response
I would like to express my appreciation for your comprehensive and detailed work. Your paper effectively navigates the journey from identifying a genetic variant to making healthcare decisions that enhance the quality of life for patients and their families. It provides valuable insights into the factors influencing decision-making, the impact of gene variants on cardiac function, and the associated risks and prognoses for individual patients. While I commend your work, I also believe that there is room for improvement through some necessary adjustments and suggestions.
One primary concern in the manuscript is the inconsistent use of acronyms. Some acronyms are defined multiple times in the paper, while others are defined unnecessarily. Additionally, certain paragraphs are complex and require greater clarity or better alignment of ideas. Lastly, I observed a lack of citations in critical sections that demand proper sourcing of information.
Response:
We would like to thank the reviewer for his/her detailed comments and for all the suggestions. We think that they allowed to greatly improve our manuscript.
This is our point-to-point reply. Please, consider that English language has been extensively reviewed by a native English speaker as suggested by the Editor.
In the following paragraphs, I will provide a detailed list of the required adjustments:
- Line 37: "heart failure" should be initially defined as HF. Afterward, you can use the acronym consistently throughout the paper.
Response:
We added the acronym for heart failure initially and throughout the text.
- Lines 25 to 29: Citations are required for the information presented in these lines.
Response:
We added references related to the information present in those lines
- Line 88: The term "the proband" is introduced for the first time. Please clarify the meaning of this term in the context of genetic testing.
Response:
We specified the meaning of proband in the context of genetic testing.
- Line 101: The acronym "WES" is not defined, and it should be introduced upon its first use.
Response:
We defined the acronym WES, as requested.
- Lines 106: The meaning of "Variant’s boundary" (the breakpoint) is not clearly defined.
Response:
We decided to remove that part as it is not essential for the understanding of the following lines.
- Lines 104 to 129: These lines discuss sequencing techniques. I recommend creating a concise figure summarizing the characteristics, advantages, and disadvantages of each technique for clarity.
Response:
Thank you for the suggestion. We added a table summarizing characteristics, advantages and disadvantages of each technique.
- Lines 123 to 124: The phrase, "It can perform a single wet-lab workflow and update the analysis to incorporate new knowledge without regenerating data [40,41]," lacks clarity and doesn't contribute significantly to the message. Please clarify its meaning.
Response:
we reformulated the sentence and clarified the concept.
- Line 139: The term "survey method" needs clarification.
Response:
We clarified the meaning using a more suitable synonym.
- Lines 201 to 208: This paragraph appears complex and would benefit from a clearer message. I recommend rephrasing it. Additionally, the phrase "to name a few" has an informal tone and should be omitted.
Response:
we have reworded the paragraph more clearly and we omitted the phrases “to name a few”.
- Line 228: It is advisable to define the acronym "VUS" in the title since it is used multiple times in this section.
Response:
In accordance with your request, we have included the acronym VUS in the title.
- Line 273 to 274: The phrase, "VUSs should not be used for cascade testing and are rarely actionable in the proband [57]," lacks clarity. Please rephrase it for better understanding.
Response:
In accordance with the request, we have modified the sentence to improve the clarity
- Line 298: Please define "ICD" upon its first use.
Response:
We have specified the acronym for ICD, as requested.
- Line 302: Although "proband" is defined here, it should have been defined in its first appearance (Line 88).
Response:
We defined the term proband in the first appearance, as you requested.
- Line 209: The term "proband" is defined again, which could be redundant.
Response:
We modified accordingly.
- Line 321: The acronym "ICC" is defined here, but it is not used elsewhere in the manuscript. Therefore, I recommend reviewing the entire manuscript for acronym definitions and using them only when the concept is mentioned more than twice.
Response:
We decided to delete the acronym ICC, since it has been used only for this single definition. We deleted the acronym PPCM (used for two times) as well, and we checked out for other acronyms used less than 2 times.
- Line 322: The acronym "SCD" is defined here, even though it was mentioned earlier (Line 66).
Response:
We decided to remove the definition and leave the acronym.
- Figure 1: Consider replacing the left box, "prevention of abnormal protein expression," with "expression of a functional protein" or a similar phrase for clarity.
Response:
Thank you for the suggestion, we modified accordingly.
- Lines 371 to 376: Citations are required for the information presented in these lines.
Response:
We specified the citations related to the information, as requested.
- Line 382: Define "AAV9," but also explain the relevance of using the AAV technique and the significance of different serotypes. This clarification would be particularly helpful in understanding Table 1, which lists several AAV serotypes.
Response:
we explained AAVs and the use of serotypes
- Lines 429 to 436: The citation [10] does not appear to reference DCM, and there are no references for the rest of this paragraph. Please review the citations throughout the article.
Response:
We noticed that during the revision phase, a zero was accidentally deleted from the citation, hence it refers to citation n.100. Additionally, we have decided to remove the sentence regarding other potential etiologies of DMC. We have reviewed all citations, as requested.
- Line 438: There is a typographical error; "PLN gene" appears as "PNL."
Lines 452 to 453: The phrase, "those with a genetic variant (genotype-positive, phenotype-negative)," is somewhat complex and can be clarified by equating it to the concept of a "carrier."
Response:
We modified the mistake in the PNL gene.
We have modified the concept between the parentheses and included the word 'carrier’.
- Lines 455 to 466: Citations are needed here. Additionally, "heart failure" should be referred to as HF.
Response:
We referred to heart failure as HF and we added the citations related to this short paragraph.
- Line 463 to 464: It is unclear whether the parents mentioned in the sentence have a genetic variant. Please provide clarification.
Response:
We clarified the concept by adding this sentence and the related citation: The study did not explicitly indicate whether patients with HF possessed specific genetic variants. However, a clear hypothesis emerged, suggesting that the inheritance of certain genetic factors (those linked to maladaptive reactions to environmental or biological stressors) implies a causal relationship between these genetic elements and the disease's progression [112].
- Line 468: The term "Genetic Variant" is capitalized, which seems inconsistent.
Response:
We rewrote the term 'Genetic Variant' in lowercase initials.
- Line 492: There is a duplication of the definition of "ARVC."
Response:
We have removed the definition and left the acronym for ARVC
- Line 498: There is a duplication of the definition of "RCM."
Response:
We have removed the definition and left the acronym for RCM
- Line 498 to 499: The phrase, "TNNC1-positive probands with mutations in the TNNC1 gene," appears redundant.
Response:
We deleted the redundant part of the phrase and we kept the first part.
- Line 510: Please define "CMR" upon its first use.
Response:
We defined CMR, as requested.
- Line 522: Use "CMR" when referring to magnetic resonance imaging (MRI).
Response:
We modified the acronym CMR instead of cardiac MRI, as requested.
- Lines 526 to 537: Citations are required for the information presented in these lines.
Response: We added the citations relative to this paragraph (4, 100, 109, 110).
- Line 547: You have defined "SCD" again.
Response:
We removed the definition and retained the acronym
- Line 549: You have defined "ICD" again.
Response:
We removed the definition and retained the acronym
- Line 566: The acronym "LGE" is not defined. Consider whether it is necessary to use the acronym here.
Response:
We explained the acronym with the definition of LGE.
- Lines 571 to 572: Please define "VA" and "AV," if necessary, as they are not commonly used across the manuscript.
Response:
We defined both VA e AV and we kept the acronym for the following use.
- Table 2: Clarify how the risk classification was defined. Incorporate citations in the table to provide the basis for this classification.
Response:
We incorporated the citations in the table, gene by gene, adding some new references as well. We explained in the text how we prepared the table using the citations and the evidence included.
- Lines 616 to 619: Consider using "SCD" for consistency. The text in this section is complex; aim for simplicity and clarity.
Response:
Thank you for your observation. We changed 'sudden cardiac death' to 'SCD.
- Line 622: The definition of "PPCM" appears unnecessary since it is mentioned only once again.
Response:
We deleted the acronym PPCM and we decided to keep only the definition, since it’s used only for two times.
- Lines 627 to 630: This paragraph lacks clarity. Please rephrase it to enhance understanding.
Response:
We have modified the sentence to improve clarity and enhance understanding:
Thus, it becomes essential to embrace the idea of individual genetic predispositions. This suggests that specific genetic variants might have a direct correlation with either favorable or unfavorable outcomes. This is even applicable to less prevalent cardiomyopathies, where previously only external or secondary causes were considered as potential etiologies [138].
- Line 642: Suggest removing the word "tries."
Response:
We removed the word tries and we kept “figure 2 summarizes …”
- Figure 2: The color of the box labeled "Maladaptive response" may not provide sufficient contrast for the arrows behind it. Please use a color with a higher contrast.
Response:
Thank you. We modified as suggested.
- Line 651: Use "SCD" for consistency.
Response:
We replaced 'sudden cardiac death' with the acronym 'SCD’.
- Line 698: You have defined "SCD" again.
Response:
We deleted 'sudden cardiac death' and left only the acronym 'SCD’.
- Line 712: The acronym "VUS" is defined again.
Response:
We deleted the definition “variant of uncertain significance” and left the acronym VUS.
- Line 713: I recommend using "HF" consistently after defining "heart failure" in its first appearance.
Response:
After the first appearance of Heart failure in the text we replaced every 'heart failure' with the acronym 'HF'. We left “heart failure” in the descriptions of the figures and in the titles of the paragraphs.
- Lines 718, 723, 726, 737: Please use "HF" for "heart failure" consistently throughout the text after its initial definition.
Response:
We have replaced 'heart failure' in the text with the acronym 'HF'. We kept heart failure for the descriptions of the figures and in the titles of the paragraphs.
- Line 744: The acronym "VUS" is defined again.
Response:
We deleted the definition of VUS and we left only the acronym VUS.
Reviewer 2 Report
The authors present an extensive, valuable, and up-to-date review of the role of genetics in the management of heart failure patients, focusing on etiology, risk evaluation, screening, and potential role in therapeutic clinical management. Also, they emphasize the importance of a multidisciplinary team approach in this field. The article includes useful summary tables, catchy diagrams, and flowcharts. I believe that the article will be of notable interest to the readers and has considerable citation potential.
I have spotted one misinterpreted statement from the cited reference that definitely needs correction, along with a few misspelled abbreviations and punctations that needed correction for better readability.
CORRECTIONS
1.
Statement in the lines 558-559 is missinterpretation of the reference No 4 (and related calculator – reference No 125) and should definitely be corrected. That needs full revision of the related paragraph containing lines 556 – 564. Actually, you wrote that “primary prevention ICD implantation should be considered with LVEF thresholds higher than 35%”, however, the correct one - based on the reference (4) and related calculator (124) - the correct text should be such as: “Decision on ICD Implantation as primary prevention in patients with LMNA related cardiomyopathies can be based on prediction score by the LMNA-risk VTA calculator assessing the risk of 5-Year Life-Threatening VTA (125), offering better risk prediction compared to the current standard of care (4) defined as ≥ 2 of the following risk factors: male sex, nonmissense mutations, nonsustained ventricular tachycardia, and a left ventricular ejection fraction (LVEF) <45%.”
Please rewrite and recompose the paragraph accordingly.
2.
Explain the abbreviation VTA (Ventricular TachyArrythnia), line 561 and elsewhere
3.
Correct the abbreviation LMVA to LMNA, line 561
4.
Change the punctuation in the sentence lines 558-561 for better readability:
From: In fact, while in secondary prevention, the implantation of a cardioverter defibrillator (ICD) is recommended in patients who have not experienced sustained symptomatic ventricular arrhythmias, the indication of ICD remains a challenging aspect of clinical care.
To: In fact, while in secondary prevention the implantation of a cardioverter defibrillator (ICD) is recommended, in patients who have not experienced sustained symptomatic ventricular arrhythmias the indication of ICD remains a challenging aspect of clinical care.
Author Response
The authors present an extensive, valuable, and up-to-date review of the role of genetics in the management of heart failure patients, focusing on etiology, risk evaluation, screening, and potential role in therapeutic clinical management. Also, they emphasize the importance of a multidisciplinary team approach in this field. The article includes useful summary tables, catchy diagrams, and flowcharts. I believe that the article will be of notable interest to the readers and has considerable citation potential.
I have spotted one misinterpreted statement from the cited reference that definitely needs correction, along with a few misspelled abbreviations and punctations that needed correction for better readability.
Response:
We would like to thank the reviewer for his/her comments. This is our point-to-point reply.
Please, consider that English language has been extensively reviewed by a native English speaker as suggested by the Editor.
- Statement in the lines 558-559 is missinterpretation of the reference No 4 (and related calculator – reference No 125) and should definitely be corrected. That needs full revision of the related paragraph containing lines 556 – 564. Actually, you wrote that “primary prevention ICD implantation should be considered with LVEF thresholds higher than 35%”, however, the correct one - based on the reference (4) and related calculator (124) - the correct text should be such as: “Decision on ICD Implantation as primary prevention in patients with LMNA related cardiomyopathies can be based on prediction score by the LMNA-risk VTA calculator assessing the risk of 5-Year Life-Threatening VTA (125), offering better risk prediction compared to the current standard of care (4) defined as ≥ 2 of the following risk factors: male sex, nonmissense mutations, nonsustained ventricular tachycardia, and a left ventricular ejection fraction (LVEF) <45%.” Please rewrite and recompose the paragraph accordingly.
Response:
We modified the lines as requested for clarity and better understanding.
- Explain the abbreviation VTA (Ventricular TachyArrythnia), line 561 and elsewhere
Response:
We explained the acronym VTA for better understanding. We kept the acronym VTA in the following uses of VTA in the same phrase.
- Correct the abbreviation LMVA to LMNA, line 561
Response:
We replaced LMVA with LMNA and we checked for other similar mistakes throughout the text.
- Change the punctuation in the sentence lines 558-561 for better readability:
From: In fact, while in secondary prevention, the implantation of a cardioverter defibrillator (ICD) is recommended in patients who have not experienced sustained symptomatic ventricular arrhythmias, the indication of ICD remains a challenging aspect of clinical care.
To: In fact, while in secondary prevention the implantation of a cardioverter defibrillator (ICD) is recommended, in patients who have not experienced sustained symptomatic ventricular arrhythmias the indication of ICD remains a challenging aspect of clinical care.
Response:
We modified the punctuation in the sentence you indicated, as requested, for better clarity.